# Systematic Review of Recommendation Systems for Course Selection

**Shrooq Algarni \* and Frederick Sheldon** 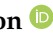

Department of Computer Science, University of Idaho, Moscow, ID 83843, USA; sheldon@uidaho.edu
\* Correspondence: alga8642@vandals.uidaho.edu

**Abstract:** Course recommender systems play an increasingly pivotal role in the educational landscape, driving personalization and informed decision-making for students. However, these systems face significant challenges, including managing a large and dynamic decision space and addressing the cold start problem for new students. This article endeavors to provide a comprehensive review and background to fully understand recent research on course recommender systems and their impact on learning. We present a detailed summary of empirical data supporting the use of these systems in educational strategic planning. We examined case studies conducted over the previous six years (2017–2022), with a focus on 35 key studies selected from 1938 academic papers found using the CADIMA tool. This systematic literature review (SLR) assesses various recommender system methodologies used to suggest course selection tracks, aiming to determine the most effective evidence-based approach.

**Keywords:** academic advising systems; course recommender systems; collaborative filtering; content-based filtering; hybrid recommender system

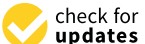



## 1. Introduction

Course recommender systems are considered an application of academic advising systems (AAS). There is currently a need to make sure that students use the information at their disposal to their best advantage in making more informed decisions regarding their academic plans [1]. This is especially important considering the advent of flexible curriculum systems in many educational institutions and the availability of an ever-widening range of courses and programs. In addition to the necessary courses that every student must take, educational institutions often offer a myriad of optional courses (e.g., tech electives). However, most students are unaware of the goals and substance of these courses; thus, they often could have chosen electives better aligned with their academic plans. Another, facet that plays a significant part in this process would be availability which often causes conflicts. Additionally, because students are becoming more numerous and diverse (in terms of their backgrounds, expertise, and ambitions), it is crucial to customize learning and advising procedures more personalized on a student -by-student basis, so that all of these factors can be considered on a more individualized basis tailored to each individuals needs and goals. Moreover, it is certainly doubtful that one learning pathway (i.e., track) would best serve them all [2,3].

Guidance counselors are often employed by educational institutions; they are responsible for assisting students in making their academic decisions. Developmental advising, prescriptive advising, and invasive advising are the three main forms of advising, and each is influenced by the objectives formulated within the advisor–student relationship. This support is focused on the development of an educational partnership between students and their academic advisers in all methods [4]. Advisors help students by assisting them with understanding the university's educational requirements, assisting them in scheduling the most appropriate modules including prerequisites, introducing them to

relevant resources, encouraging leadership and campus involvement, assisting with career development, ensuring that they finish their studies on time, and assisting them in finding ways to make their educational experience personally relevant [5]. *However, the counselors are frequently overburdened with too many students and not enough time.* Some students can become dissatisfied with the kind of academic guidance the counselors offer. When it comes to comprehending, organizing, and putting ideas for academic achievement into practice, excellent advising produces positive results, whereas poor advising frustrates students and can even be detrimental to their development [6,7] and ultimate success.

A software solution that can manage the advice process effectively and efficiently is needed to assist the educational process and to relieve the educational institutions' players. A course recommender system can act as a strategic partner in the process of aiding the student (and advisor) in achieving their educational goals and supporting and encouraging their study plan [6,8]. However, unlike most other existing recommendation systems, course recommender systems (CRS) must deal with a sizable decision space that multiplies combinatorically with the number of courses; programs; and the various backgrounds, skills, and goals of a student while simultaneously being subject to *numerous* restrictions (e.g., maximum credit hour load, course prerequisites, sequencing etc.) [1].

To understand the preferences of various users and forecast products that correspond to their demands, recommender systems scour large databases for important patterns. The word "item" in this context refers to any course, educational component, book, service, application, or product. Machine-learning and data-mining methods are mostly used by CRSs to sort all these constraints toward accomplishing each students' goals and objectives. Moreover, these same methods are widely utilized in e-commerce and by shops to increase their sales and viewership. These days, those same techniques are being used more frequently for educational recommendation and advising purposes [9], making the whole student-advisor process more effective, efficient and clear cut; win/win for both the institution and its constitutes.

Personalized recommendation systems (PRS) are becoming more and more common in a variety of industries, such as e-commerce, music and video streaming, and they are now making their way into the educational space [3]. These systems strive to make recommendations that are uniquely suited to each user's tastes and preferences, which makes them extremely pertinent in the context of course recommendation.

The "cold start problem" is a challenge that CRSs encounter. This issue emerges when new students enroll in a program while the CRS lacks sufficient data on them to provide reliable recommendations. Different approaches have been developed to address this issue, which recommender systems often confront in a variety of different industries [10].

A substantial contribution to the field of CRSs is made herein by our analysis. This is accomplished by offering a comprehensive systematic review of the literature (SLR) of empirical research in the field and highlighting the most efficient approaches supported by experiential data. We identify and highlight the knowledge gaps and potential constraints, prompting the community to direct future research endeavors accordingly. Additionally, we explore the difficulties faced by CRSs, particularly those related to handling sizable decision spaces and the cold start problem, providing insightful information about these intricate topics. This study also provides ideas for improving CRSs, considering the growing need for individualized recommendations for fruitfully navigating the complexities of an academic curriculum.

## 2. Motivation and Rationale of the Research

The need for empirical evidence to validate theoretical frameworks that then can be accepted by the scientific community serves as the driving force behind this survey. To the best of our knowledge, no systematic reviews concerning empirical studies in this specific topic exist after a search of the pertinent literature covering the previous six years. As a result, it was necessary to provide the community with an authoritative summary. The purpose of this research therefore is to close or at least lesson that gap.

Any given study's worth stems from both its inherent qualities and how it complements and advances earlier works. Thus, collecting all the objectives and reliable findings from earlier studies would be a step toward grasping the big picture and forming a roadmap of our consequent knowledge within the field. In a way, the goal of our study was to organize the voluminous quantity of publications by critically examining, assessing, and synthesizing earlier empirical findings.

The absence of empirical validation forms the additional value of research in the field of CRSs in the body of current literature and serves as a valuable result from this study. We combed through the massive body of literature, critically analyzing and evaluating earlier empirical findings to present a comprehensive overview of the existent strategies employed thus far. We draw attention to the positive and negative aspects of earlier research, point out any potential drawbacks, and motivate the research community to rephrase or rethink pertinent study questions and/or hypotheses. The lack of empirical evidence in the existing literature on the added value of research in the specific area of CRSs inspired us to dig deeper. We aimed to sift through the vast body of publications, critically examining, assessing, and synthesizing previous empirical findings to provide an exhaustive account of the applied research approaches used thus far. We highlight the successes and shortcomings of previous studies, identify potential limitations, and seek to inspire the research community to (re)consider these conclusions in designing future studies.

The creation of efficient course recommender systems is necessary due to the expanding complexity of curricula offered and the increased need for individualized learning experiences in educational institutions. While general recommender systems have received a great deal of research, there are not many thorough studies that concentrate solely on course recommender systems. This results from the difficulties this field presents, such as managing prerequisite specifications and a developing course catalog per student growth and shifting educational objectives.

## 3. Research Questions

This section will discuss the fundamental questions to be investigated in the study. Two types of research questions must be answered:

### 3.1. Questions about the Used Algorithms

- What preprocessing methods were applied?
- What recommendation system algorithms were used in the paper?
- What are the applied evaluation metrics?
- What are the performance results of applied evaluation metrics?

### 3.2. Questions about the Used Dataset

- Is the dataset published or accessible?
- How many records are there in the dataset?
- How many unique student records are there in the dataset?
- How many unique course records are there in the dataset?
- How many features are there in the dataset?
- How many features are used from the existing features?
- How many unique majors are there in the dataset?
- How did the authors split the training and testing set?

### 3.3. Questions about the Research

- What is the type of comparative produced in the study (algorithm level, preprocessing level, or data level)?
- What is the main aim of the study?
- What are the strong points of the research?
- What are the weak points of the research?

## 4. Research Methodology

Figure 1 summarizes our method for locating, including, or eliminating documents. To broaden our coverage, we looked for pertinent papers by "snowballing," i.e., starting with a core group of papers that we believed to be in scope and extending our list of consideration based on papers that these core papers referenced. We also conducted a thorough search of available research paper databases. We followed the same general methodology as Iatrellis et al. in [1].

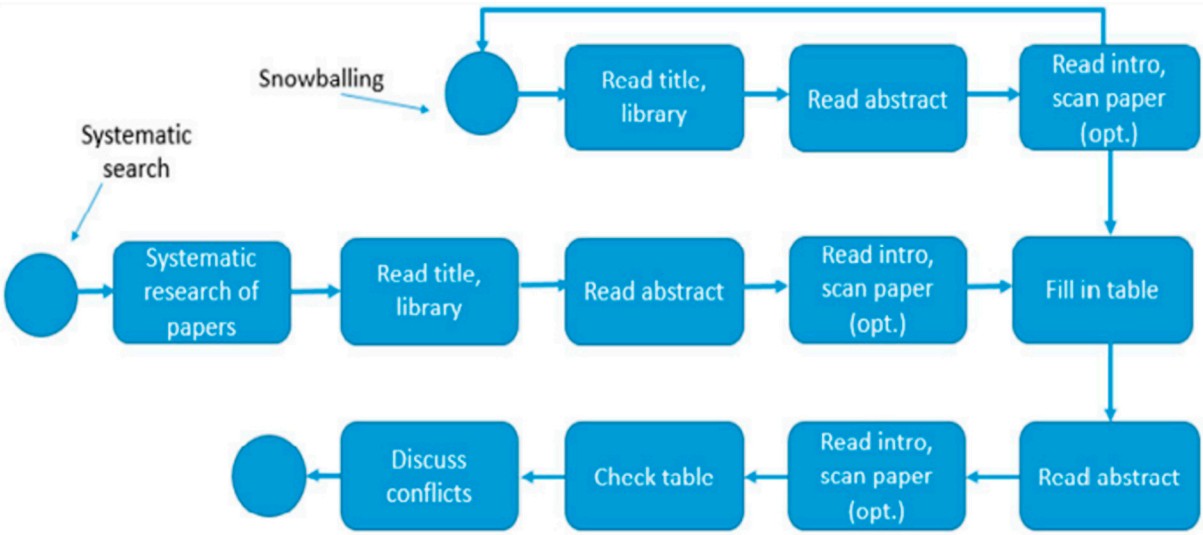

**Figure 1.** Methodology of SLR [1].

We used CADIMA [11] for the construction of this SLR. Three research databases were searched: IEEE, ACM, and Science Direct. The initial search was conducted using the string ("academic advising system" OR "Academic Advisor System") AND ("Course recommendation system" OR "Course selection system" OR "Academic advising framework"). A total of 700 research papers were included from IEEE, 757 were included from ACM, and 481 research papers were included from Science Direct, making a total of 1938 unique records that were included in this stage. However, 12 research papers were deleted using the automated removal function, making the final number of research papers 1926 in the initial search.

### 4.1. Title-Level Screening Stage

Next, we conducted a filter using a criteria list and keywords. The keywords used in the filter were: (course education, selection, recommendation system, recommender, machine learning, deep learning, algorithm, dataset, accuracy, evaluation, and data mining). The former keywords aim to help the authors in scanning the papers more efficiently and reduce paper reading time. The criteria listed in the title included two main definitions:

- The study addresses recommendation systems in the Education sector.
- The study must be primary.

In this stage, we excluded 1199 research papers; thus, the number of papers was lowered to 721 in the next stage. Examples of research papers that were excluded in this stage are included in Table 1.

**Table 1.** The reasoning behind the exclusion of some papers in the title-level screening stage.

| Author | The Study Addresses Recommendation Systems in the Education Sector | Primary Study |
|---|---|---|
| Shminan et al. [12] | No | Yes |
| Wang et al. [13] | No | Yes |
| Shaptala et al. [14] | No | Yes |
| Zhao et al. [15] | No | Yes |
| ID Wahyono et al. [16] | No | Yes |

*4.2. Abstract-Level Screening Stage*

To efficiently reduce paper reading time, the same criteria used for the title-level screening were used again in the abstract-level screening stage:

- The study addresses recommendation systems in the Education sector.
- The study must be primary.

The previous criteria were also applied in the title stage first. In this stage, 533 research papers were excluded, and 194 were included in the full-text article scanning. Examples of research papers that were excluded in this stage are included in Table 2.

**Table 2.** The reasoning behind the exclusion of some papers in the abstract-level screening stage.

| Author | The Study Addresses Recommendation Systems in the Education Sector | Primary Study |
|---|---|---|
| Elghomary et al. [17] | No | Yes |
| Mufizar et al. [18] | No | Yes |
| Sutrisno et al. [19] | No | Yes |
| Gan et al. [20] | No | Yes |
| Ivanov et al. [21] | No | Yes |

*4.3. Full-Text Article Scanning Stage*

The criteria list in the full text included the other two definitions after a quick scan of the full text of each study:

- The study was written in the English language.
- The study implies empirical experiments and provides the experiment's results.

Two research papers were excluded in this stage, making the number of studies for the next stage 192. Details about the two excluded research papers and the reasoning for their exclusion are illustrated in Table 3.

**Table 3.** The reasoning behind the exclusion of research papers in the full-text scanning stage.

| Author | Reason of Exclusion |
|---|---|
| Anupama et al. [22] | Did not imply empirical experiments and did not provide experiments results |
| Sabnis et al. [23] | The full text is not accessible |

*4.4. Full-Text Article Screening Stage*

The authors applied a quality assessment that led to the selection of 35 research passed out of the remaining 192 studies for detailed analysis by using a scoring approach that gives each study a score based on the answers to five questions. These questions are:

- Q1: Did the study conduct experiment in the course selection and courses recommendation system?

- Q2: Is there a comparison with other approaches in the conducted study?
- Q3: Were the performance measures fully defined?
- Q4: Was the method used in the study clearly described?
- Q5: Was the dataset and number of training and testing data identified?

Each study was given a score of 3 for each question: 0, 0.5, and 1—representing no, partially, and yes, respectively. Finally, a study was considered if it scores 3 or more points out of 5 from the previous questions. Table 4 includes some example research papers from this stage and elaborates how and why they were included or excluded.

**Table 4.** The reasoning behind the exclusion or inclusion of some papers in the full-text article screening stage.

| Author | Score | | | | | Total Score | Included |
|---|---|---|---|---|---|---|---|
| | Q1 | Q2 | Q3 | Q4 | Q5 | | |
| Britto et al. [24] | 1 | 0 | 0.5 | 0.5 | 0.5 | 2.5 | No |
| Obeidat et al. [9] | 0.5 | 1 | 0.5 | 0.5 | 0.5 | 3 | Yes |

*4.5. Data Extraction Stage*

In this stage, the authors aim to extract the answers to the defined questions in the research questions from each study of the 35 papers included in this SLR after reading the full text. Figure 2 shows the flow diagram depicting the study selection process.

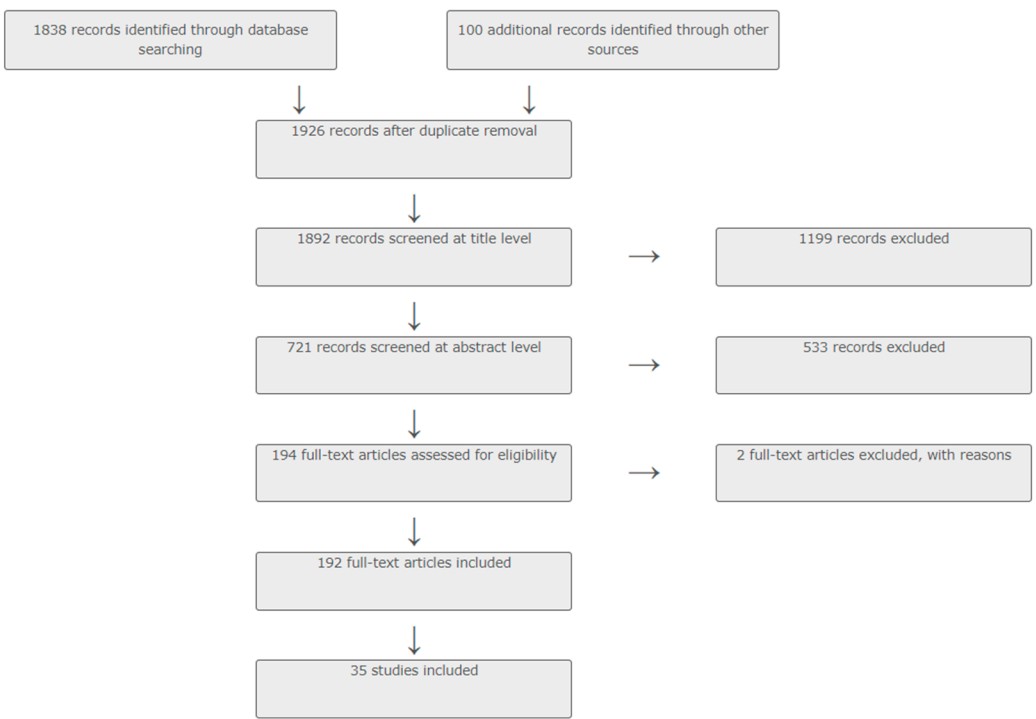

**Figure 2.** Flow diagram depicting the study selection process.

## 5. Research Results

The authors give their conclusions in this part after analyzing the published case studies. Most published case studies are exploratory or experimental studies, according to the study methodology used. Others are empirical research or surveys, while some of them are evaluative studies. The resulting SLR dataset, which contains the answers to the questions mentioned in the research questions, was analyzed, and explored using Excel and Tableau.

According to Peng [25], there are three main approaches to designing recommender systems: collaborative filtering (CF), content-based filtering (CBF), and hybrid approaches. Most publications in recommender systems revolve around the three mentioned algorithms, while some publications try to achieve better results with novel approaches or similarity-based filtering. For this SLR, the approaches used to design course recommender systems included collaborative filtering, content-based filtering, novel approaches, hybrid recommender systems, and similarity-based recommender systems.

Due to their parallelization capabilities, collaborative filtering-based approaches are frequently utilized in big data-processing systems. The actions of a set of users are used by CF recommendation systems to provide recommendations to other users [25]. CF is a technique that can filter out items that a user might like on the basis of reactions by similar users (i.e., searching a large group of people and finding a smaller set of users with tastes similar to a particular user).

The basic goal of content-based recommenders is to provide recommendations for products based on how various users or goods are similar to one another. By examining the descriptions of those products, this algorithm ascertains and distinguishes the primary characteristics of a certain user's favorite goods. The user's profile will thereafter have a record of these choices. The system then suggests products that are more comparable to the user's profile. Additionally, content-based recommendation algorithms can identify the user's individual preferences and can suggest uncommon items that are not of much interest to other users. However, this technique needs heaps of domain expertise because the feature representations of things are created manually to some extent. Additionally, a users' capacity to broaden their existing interests is constrained by the fact that content-based recommendation systems can only provide recommendations based on their current interests [25].

On the other hand, to overcome any possible shortcomings in conventional recommendation systems, hybrid-based recommendation systems integrate the benefits of several recommendation strategies. The most popular hybrid recommender system technique for preventing sparsity, enabling scalability, and dealing with cold-start issues combines content-based and knowledge-based recommendation techniques with CF recommendation techniques [25].

### 5.1. The Studies Included in the SLR

This subsection provides basic information about the research papers included in the SLR, such as the authors, year of publication, algorithms used, and the comparative types. This subsection contains five subsubsections for each recommender system approach from the five main algorithms in research studies included from the literature. The approaches are (i) collaborative filtering, (ii) content-based filtering, (iii) hybrid recommender systems, (iv) novel approaches, and (v) similarity-based filtering included in the SLR. Thus, we included thirteen collaborative filtering, two content-based filtering, six hybrid recommender systems, thirteen novel approaches, and only one similarity-based filtering study.

The SLR reveals a diverse range of recommender system approaches. These include (a) collaborative filtering, (b) content-based filtering, (c) hybrid recommender systems, (d) novel approaches, and (e) similarity-based filtering; all of which are the predominant methodologies. It is noteworthy that collaborative filtering and novel approach-based methodologies are the most common, with 13 studies each, signifying their popularity in this research domain.

The 13 collaborative filtering studies span from 2017 to 2022, with most published in 2019. These studies employ a variety of algorithms, including traditional collaborative filtering, two-stage collaborative filtering, and neural collaborative filtering, showcasing the versatility of this approach. Only two content-based filtering studies were included in the SLR, stressing this approach's relative underutilization. Interestingly, the 2021 study employed the weighted cosine similarity and TF–IDF, suggesting an evolution in the sophistication of content-based filtering methodologies.

Hybrid recommender systems, which typically combine collaborative and content-based filtering, were the focus of six studies. Some of these studies also incorporated machine-learning algorithms such as KNN and k-means clustering to boost recommendation accuracy. The novel approaches category, with 13 studies, demonstrates a propensity for researchers to experiment with newer methodologies, such as machine learning and deep learning algorithms. These studies, which utilized techniques such as SVM, KNN, random forests, linear regression, Naive Bayes, decision trees, and even LSTM, underscore the increasingly central role of machine learning in the development of course recommender systems. The lone study using similarity-based filtering, published in 2017, utilized similarity-based regularization with matrix factorization, indicating that this approach may be less explored.

Overall, the SLR points towards an increasing reliance on sophisticated machine-learning and hybrid methodologies in the field of CRS. Nevertheless, traditional collaborative filtering still maintains a strong presence, and the relative scarcity of content-based and similarity-based studies presents opportunities for future research.

### 5.1.1. Collaborative Filtering Studies

Table 5 shows the recommender system publications that used the collaborative filtering approach along with the authors, year of publication, the used algorithms, and the type of comparative. All 13 research studies found utilizing collaborative filtering were in the algorithm level type of comparative.

**Table 5.** Recommender system publications that used collaborative filtering along with the authors, year of publication, used algorithms, and comparative type.

| Authors and Year | Algorithms Used | Comparative Type |
|---|---|---|
| A. Bozyiğit et al., 2018 [26] | • Collaborative filtering;<br>• OWA (ordered weighted average). | Algorithm level |
| B. Mondal et al., 2020 [27] | • Collaborative filtering. | Algorithm level |
| E. L. Lee et al., 2017 [28] | • Two-stage collaborative filtering;<br>• Personalized Ranking Matrix Factorization (BPR-MF);<br>• Course dependency regularization;<br>• Personalized PageRank;<br>• Linear RankSVM. | Algorithm level |
| I. Malhotra et al., 2022 [29] | • Collaborative filtering;<br>• Domain-based cluster knowledge;<br>• Cosine pairwise similarity evaluation;<br>• Singular value decomposition ++;<br>• Matrix factorization. | Algorithm level |
| L. Huang et al., 2019 [30] | • Cross-user-domain collaborative filtering. | Algorithm level |
| L. Zhao et al., 2021 [31] | • Improved collaborative filtering;<br>• Historical preference fusion similarity. | Algorithm level |
| M. Ceyhan et al., 2021 [32] | • Collaborative filtering;<br>• Correlation-based similarities: (Pearson correlation coefficient, median-based robust correlation coefficient);<br>• Distance-based similarities: Manhattan and Euclidian distance similarities. | Algorithm level |
| R. Obeidat et al., 2019 [9] | • Collaborative filtering;<br>• K-means clustering;<br>• Association rules (Apriori algorithm, sequential, pattern discovery using equivalence classes algorithm). | Algorithm level |

**Table 5.** *Cont.*

| Authors and Year | Algorithms Used | Comparative Type |
|---|---|---|
| S. Dwivedi et al., 2017 [33] | • Collaborative filtering;<br>• Similarity log-likelihood. | Algorithm level |
| S.-T. Zhong et al., 2019 [34] | • Collaborative filtering;<br>• Constrained matrix factorization. | Algorithm level |
| Z. Chen et al., 2017 [35] | • Collaborative filtering;<br>• Association rules (Apriori). | Algorithm level |
| Z. Chen et al., 2020 [36] | • Collaborative filtering;<br>• Improved cosine similarity;<br>• TF–IDF (term frequency–inverse document frequency). | Algorithm level |
| Z. Ren et al., 2019 [37] | • Neural Collaborative Filtering (NCF). | Algorithm level |

Four collaborative filtering research studies were published in 2019, while three papers were published in 2017. Moreover, two papers were published in 2020, and the same in 2021. Finally, only one paper was published in 2022.

5.1.2. Content-Based Filtering Studies

Table 6 shows the two content-based filtering research studies that are included in this SLR. One publication was published in 2020 and had the preprocessing-level type of comparative, while the other was published in 2021 and had the algorithm-level type of comparative. The first content-based research published in 2020 used the basic algorithm without any modifications, while the latest content-based research published in 2021 utilized the weighted cosine similarity and TF–IDF to acquire better course recommendations.

**Table 6.** Recommender system publications that used content-based filtering along with the authors, year of publication, used algorithms, and comparative type.

| Authors and Year | Used Algorithms | Comparative Type |
|---|---|---|
| A. J. Fernández-García et al., 2020 [38] | • Content-based filtering. | Preprocessing level |
| Y. Adilaksa et al., 2021 [39] | • Content-based filtering;<br>• Weighted cosine similarity;<br>• TF–IDF. | Algorithm level |

5.1.3. Hybrid Recommender System Studies

Table 7 lists algorithms used in six hybrid recommender system papers from an SLR, published from 2018 to 2022. Each year saw one relevant paper, except 2021 with two. These papers, focusing on the algorithm-level comparison, typically employed collaborative filtering and content-based filtering, with some also using machine-learning algorithms such as KNN and k-means for better accuracy.

**Table 7.** Recommender system publications that used hybrid recommender system along with the authors, year of publication, used algorithms, and comparative type.

| Authors and Year | Used Algorithms | Comparative Type |
|---|---|---|
| Esteban, A. et al., 2020 [40] | • Hybrid recommender system;<br>• Collaborative filtering;<br>• Content-based filtering;<br>• Genetic algorithm. | Algorithm level |
| M. I. Emon et al., 2021 [41] | • Hybrid recommender system;<br>• Collaborative filtering;<br>• Association rules (Apriori algorithm). | Algorithm level |

**Table 7.** *Cont.*

| Authors and Year | Used Algorithms | Comparative Type |
|---|---|---|
| S. Alghamdi et al., 2022 [42] | ● Hybrid recommender system;<br>● Content-based filtering;<br>● Association rules (Apriori algorithm);<br>● Jaccard coefficient. | Algorithm level |
| S. G. G et al., 2021 [43] | ● Hybrid recommender system;<br>● Collaborative filtering;<br>● Content-based filtering;<br>● Lasso;<br>● KNN;<br>● Weighted average. | Algorithm level |
| S. M. Nafea et al., 2019 [44] | ● Hybrid recommender system;<br>● Felder–Silverman learning styles model;<br>● K-means clustering. | Algorithm level |
| X. Huang et al., 2018 [45] | ● Hybrid recommender system;<br>● Association rules;<br>● Improved multi-similarity. | Algorithm level |

### 5.1.4. Studies Based on Machine Learning

Table 8 shows the 13 research papers that presented novel approaches to recommend courses to students.

**Table 8.** Recommender system publications that used novel approaches along with the authors, year of publication, used algorithms, and comparative type.

| Authors and Year | Used Algorithms | Comparative Type |
|---|---|---|
| A. Baskota et al., 2018 [46] | ● Forward feature selection;<br>● K-Nearest Neighbor (KNN);<br>● Multi-class Support Vector Machines (MC-SVM). | Algorithm level |
| Jiang, Weijie et al., 2019 [47] | ● Goal-based filtering;<br>● LSTM recurrent neural network. | Algorithm level |
| Liang, Yu et al., 2019 [48] | ● Currency rules;<br>● C4.5 decision tree. | Preprocessing level |
| M. Isma'il et al., 2020 [49] | ● Support Vector Machine (SVM). | Algorithm level |
| M. Revathy et al., 2022 [50] | ● KNN-SMOTE. | Algorithm level |
| Oreshin et al., 2020 [51] | ● Latent Dirichlet Allocation;<br>● FastTextSocialNetworkModel;<br>● Catboost. | Algorithm level |
| R. Verma et al., 2018 [52] | ● Support Vector Machines;<br>● Artificial Neural Networks (ANN). | Algorithm level |
| S. D. A. Bujang et al., 2021 [53] | ● Random forests. | ● Algorithm level<br>● Preprocessing level |
| S. Srivastava et al., 2018 [54] | ● Support Vector Machines with radial basis kernel;<br>● KNN. | Algorithm level |
| T. Abed et al., 2020 [55] | ● Naive Bayes. | Algorithm level |
| V. L. Uskov et al., 2019 [56] | ● Linear regression. | Algorithm level |
| V. Sankhe et al., 2020 [57] | ● Skill-based filtering;<br>● C-means fuzzy clustering;<br>● Weighted mode. | Algorithm level |
| V. Z. Kamila et al., 2019 [58] | ● KNN;<br>● Naive Bayes. | algorithm level |

A total of 11 research papers focused on the algorithm-level type of comparative, while only one research was on the preprocessing-level type of comparative. Additionally, one study [53] addressed two types of comparatives: algorithm level and preprocessing level. Novel approaches for CRS started in 2018, with a relatively large number of publications in the year (three research papers), followed by four publications in 2019 and the same number of publications in 2020. Then, the number decreased in the last two years to only one research each year.

Novel approaches focused mainly on the utilization of machine-learning algorithms including SVM, KNN, random forests, linear regression, Naive Bayes, and decision trees for the task of course recommendations. Moreover, two papers utilized deep learning for the task by using ANNs in [52] or using LSTM in [47].

5.1.5. Similarity-Based Study

Table 9 shows the only research paper that utilized similarity-based filtering for the task of course recommendations. This research paper was published in 2017 and focused on the algorithm level. The authors utilized similarity-based regularization with matrix factorization to improve course recommendations.

**Table 9.** Recommender system publication that used similarity-based filtering along with the authors, year of publication, used algorithms, and comparative type.

| Authors and Year | Used Algorithms | Comparative Type |
|---|---|---|
| D. Shah et al., 2017 [59] | • Similarity-based regularization; <br> • Matrix factorization. | Algorithm level |

## 6. Key Studies Analysis

Figure 3 reveals that recommender system publications over the past six years have primarily used collaborative filtering and novel approaches, each with a total of 13 research projects. The surge in novel approaches between 2018 and 2020 indicates a shift from traditional methods. However, content-based filtering and similarity-based regularization were used less frequently, with only two and one research papers, respectively.

Collaborative filtering methods, as used in the 13 studies, varied from addressing course repetition issues and improving accuracy and recall rates to offering big data recommendations and course score predictions.

Novel approaches, too, demonstrated diversity in their methods, with systems based on Support Vector Machine and K-Nearest Neighbor techniques, recurrent neural networks, decision trees, and other machine learning algorithms. Content-based filtering methods aimed at solving the student dropout problem and improving recommendation accuracy, whereas the sole similarity-based filtering study extended the online matrix factorization technique for its' CRS.

These findings demonstrate the dynamic and evolving nature of recommender system research, with a clear emphasis on improving the accuracy and efficacy of course recommendations. Despite the dominance of collaborative filtering and novel approaches, the presence of content-based and similarity-based filtering methods illustrates the wide range of techniques used in the field. As such, future research could explore a broader application of these less utilized methods to diversify the solutions available in the educational context.

In the realm of Collaborative Filtering (CF), the 13 studies examined each addressed the unique facets of course recommendation. For instance, some studies focused on the overlooked issue of course repetition, while others aimed at improving accuracy and recall rates. Certain studies harnessed machine-learning strategies to suggest appropriate courses to learners based on their prior performance and learning history. The wide range of applications of collaborative filtering, from the prediction of course scores to big data recommendations, highlights its versatility in this field.

## Types of recommender system publications per year

**Figure 3.** Number of each type of recommender system publication for the previous 6 years.

Content-based filtering, though less frequently utilized, offered innovative solutions to significant issues such as student dropout and recommendation accuracy. For example, one study aimed to reverse the dropout problem by recommending courses that would increase graduation rates.

Hybrid Recommender Systems were also explored in several studies, aiming to blend the strengths of different approaches to overcome inherent limitations. These systems tackled a variety of challenges, from assisting college students in selecting electives to improving the accuracy of recommendations by considering students' implicit interests and learning styles.

The biggest variety of methods was found in the Novel Approaches category, where they included Support Vector Machines, K-Nearest Neighbor methods, Recurrent Neural Networks, and Decision Trees. These methods were frequently employed to advise students on the most alluring graduate programs or to get them ready for specific courses they were interested in. They also concentrated on forecasting students' academic results and early detection of potential dropout causes.

Finally, the one study that used similarity-based filtering expanded the use of online matrix factorization for its' CRS, demonstrating the method's potential for use in subsequent studies. Overall, the variety and depth of objectives and contributions found in the extant research works show how recommender system research is dynamic and ever evolving.

Each distinct method offers a new perspective to enhance the precision and effectiveness of course recommendations. This broad range of options offers a solid framework for further study, and promising advancements in individualized learning.

*6.1. Discussion of Aims and Contributions of the Existing Research Works*

This subsection discusses the research aim of each study in the SLR depending on the recommender system algorithm used.

6.1.1. Aim of Studies That Used Collaborative Filtering

(1)    Authors in [26] used ordered weighted average (OWA) to address the problem that most other studies make recommendations based on the student's previous academic performance. None of the current studies take course repetition into account when calculating student performance; instead, they only look at the most recent grades on the transcript for repeated courses. They made the premise that students' final grades in a course may not accurately reflect their performance because they may retake a course multiple times to improve their performance. Therefore, using the student's most recent grades alone may not result in the best suggestions,

(2)    Authors in [27] offered a machine-learning strategy to suggest appropriate courses to learners based on their prior performance and learning history,

(3)    Authors in [28] enabled the task of course recommendation to be handled using a CF-based model. They list many obstacles to using the current CF models to create a course recommendation engine, such as the absence of ratings and metadata, the uneven distribution of course registrations, and the requirement for course dependency modeling,

(4)    The system suggested by Malhorta et al. [29] will assist students in enrolling in the finest optional courses according to their areas of interest. This method groups students into clusters according to their areas of interest, then utilizes the matrix factorization approach to analyze past performance data of students in those areas to forecast the courses that a specific student in the cluster can enroll in,

(5)    Authors in [30] proposed the CUDCF (Cross-User-Domain Collaborative Filtering) algorithm, which uses the course score distribution of the most comparable senior students to precisely estimate each student's score in the optional courses,

(6)    The main aim of the authors in [31] was to improve the precision and recall rate of recommendation results by improving the collaborative filtering algorithm,

(7)    Students' grade prediction using user-based collaborative filtering was introduced by authors in [32],

(8)    Authors in [9] improved association rule generation and coverage by clustering,

(9)    Authors in [33] suggested utilizing big data recommendations in education. According to the student's grades in other topics, this study uses collaborative filtering-based recommendation approaches to suggest elective courses to them,

(10)   To forecast sophomores' elective course scores, authors in [34] presented the Constrained Matrix Factorization (ConMF) algorithm, which can not only assist students in choosing the appropriate courses but also make the most efficient use of the scarce teaching resources available at universities,

(11)   Authors in [35] applied the interestingness measure threshold and association rule of data-mining technology to the course recommendation system,

(12)   Authors in [36] improved the accuracy of recommendations by using the improved cosine similarity,

(13)   Neural Collaborative Filtering (NCF), a deep learning-based recommender system approach, was presented by authors in [37] to make grade predictions for students enrolled in upcoming courses.

6.1.2. Aim of Studies That Used Content-Based Filtering

(1) Authors in [38] introduce a method that does more than just estimate dropout risk or student performance; it also takes action to support both students and educational institutions, which helps to reverse the dropout problem. The goal is to increase graduation rates by creating a recommender system to help students choose their courses,

(2) Authors in [39] improve the accuracy of recommendations by using weighted cosine similarity instead of traditional cosine similarity.

6.1.3. Aim of Studies That Used Hybrid Recommender Systems

(1) College students were assisted in selecting electives by combining a multi-criteria hybrid recommendation system that utilizes CF and CBF with genetic optimization, which was introduced by authors in [40],

(2) Authors in [41] overcame the performance implications of traditional algorithms by presenting a hybrid approach that includes using association rule-mining and collaborative filtering,

(3) Authors in [42] used data mining and recommendation systems to assist academic advisers and students in creating effective study plans, particularly when a student has failed a few courses,

(4) Authors in [43] overcame the cold-start drawback of collaborative filtering and the domain knowledge requirement of content-based filtering by using a hybrid approach that combines both,

(5) Authors in [44] represented the student learning styles and the learning object profiles using the Felder–Silverman learning styles model, thus improving the overall accuracy of recommendations,

(6) Authors in [45] suggested a Course Recommendation Model in Academic Social Networks Based on Association Rules and Multi-similarity (CRM-ARMS) that is based on academic social networks, a hybrid approach combining an association rules algorithm and an improved multi-similarity algorithm of multi-source information, which can recommend courses by possible relationships between courses and the user's implicit interests.

6.1.4. Aim of Studies That Used Novel Approaches

(1) Authors in [46] provided a recommendation system based on the Support Vector Machine and K-Nearest Neighbor techniques that suggest the most appealing graduate programs to students,

(2) Authors in [47] helped students prepare for target courses of interest, they created a novel recommendation system based on recurrent neural networks that are tailored to each student's estimated previous knowledge background and zone of proximal development,

(3) The decision tree's upgraded algorithm is applied by the authors in [48] to the data on college electives in recent years after the currency rules and C4.5 algorithm are coupled to extract the statute rules from the student elective database,

(4) Authors in [49] built an autonomous course recommender system for undergraduates using five classification models: linear regression, Naive Bayes, Support Vector Machines, K-Nearest Neighbor, and decision tree algorithm,

(5) By applying feature selection and extraction approaches to reduce the dimensionality, early detection of dropout factors is made possible by the authors in [50]. Unbalanced data may occur during feature extraction, which could have an impact on the usefulness of machine-learning approaches. To manage the oversampling of unbalanced data and create a balanced dataset, the Synthetic Minority Oversampling Technique is then used with Principal Compound Analysis,

(6) Students' academic outcomes prediction using a machine-learning approach that utilizes the Catboost algorithm was proposed in [51],

(7) An effective method has been put out in [52] that makes use of SVM to guarantee academic success in optional courses through its predictions and to maintain student topic preferences for the better attainment of bilateral academic quality learning outcomes,

(8) Authors in [53], to reduce overfitting and misclassification results by imbalanced multi-classification based on oversampling Synthetic Minority Oversampling Technique (SMOTE) using two feature selection methods, suggested a multiclass prediction model. According to the results, the presented model integrates with RF and significantly improves with an f-score of 99.5%,

(9) Research [54] conducted for choosing open elective courses at a prestigious private university is highlighted. The classification techniques KNN and Support Vector Machine with Radial Basis Kernel are reviewed, used, and compared during the data-mining process. Additionally, the article seeks to replace the current heuristic process's mathematical underpinning with data mining techniques,

(10) Authors in [55] improved the accuracy of course recommendations using a machine-learning approach that utilizes the Naive Bayes algorithm,

(11) Authors in [56] developed a list of suggestions for academics and professionals on the choice, configuration, and application of ML algorithms in predictive analytics in STEM education,

(12) Based on a variety of criteria, authors in [57] mapped their present-day students to their alumni students. Then, in contrast to earlier articles that employed k-means, they used c-means and fuzzy clustering to find a superior way to predict the student's elective course,

(13) The goals of the study [58] were to determine how KNN and Naive Bayes can be used to suggest the best and most advanced course options for students.

### 6.1.5. Aim of Studies That Used Similarity-Based Filtering

The only study that discussed using similarity-based filtering for AAS was introduced by the authors in [59], where they presented a machine-learning approach for course recommender systems by extending the online matrix factorization technique.

### 6.2. Description of Datasets Used in the Studies

This subsection will discuss the dataset description for the research studies included in the SLR divided by the recommender system algorithm used in each research and including whether the dataset used was public or private, and the numbers of records, students, courses, majors, features, and used features in the dataset. Additionally, any preprocessing steps mentioned in the research were included in Table 10. Finally, the data-splitting method was documented, if available.

**Table 10.** Description of datasets used in each collaborative filtering study, the preprocessing steps, and the data-splitting method.

| Authors and Year | Public | Records | Students | Courses | Majors | Features | Used Features | Preprocessing Steps | Data-Splitting Method |
|---|---|---|---|---|---|---|---|---|---|
| A. Bozyiğit et al., 2018 [26] | No | N/A | 221 | 76 | N/A | N/A | N/A | N/A | Ten-fold cross-validation. |
| B. Mondal et al., 2020 [27] | No | 300 | 300 | N/A | N/A | 48 | 12 | ● Data cleaning: lowercase conversion, removing punctuation, striping white spaces. | N/A |

**Table 10.** *Cont.*

| Authors and Year | Public | Records | Students | Courses | Majors | Features | Used Features | Preprocessing Steps | Data-Splitting Method |
|---|---|---|---|---|---|---|---|---|---|
| E. L. Lee et al., 2017 [28] | No | 896,616 | 13,977 | N/A | N/A | N/A | N/A | • Ignore the students whose 4-year registration records are incomplete. | Nested time-series split cross-validation (class 2008, class 2009 as a training set, and class 2010 as a testing set). |
| I. Malhotra et al., 2022 [29] | No | N/A | 1780 | N/A | 9 | N/A | N/A | N/A | N/A |
| L. Huang et al., 2019 [30] | No | 52,311 | 1166 | N/A | 8 | N/A | N/A | N/A | N/A |
| L. Zhao et al., 2021 [31] | No | N/A | 43,916 | 240 | N/A | N/A | N/A | • Group data based on interest data points, • Eliminate noise by filtering the data noise constrained in 0,1, • Normalize all numerical features. | Five-fold cross-validation. |
| M. Ceyhan et al., 2021 [32] | No | N/A | 1506 | 1460 | N/A | N/A | N/A | • The updated grade is taken into consideration if a student retakes any course. | • Nested time-series split cross-validation, • Train = 91.7% (from 2010/11-F to 2019/20-S), • Test = 8.3% (the whole 2020/21-F). |
| R. Obeidat et al., 2019 [9] | Yes | 22,144 | 10,000 | 16 | N/A | N/A | N/A | • Remove incomplete records • Calculate the order of courses sequences events for each student, • Convert grades to a new grade scale, • Cluster students. | N/A |
| S. Dwivedi et al., 2017 [33] | No | N/A | N/A | N/A | N/A | N/A | N/A | • Data cleaning, • Data discretization (converting low-level concept to high-level concept). | N/A |
| S. -T. Zhong et al., 2019 [34] | No | N/A | N/A | N/A | 8 | N/A | N/A | N/A | N/A |

**Table 10.** *Cont.*

| Authors and Year | Public | Records | Students | Courses | Majors | Features | Used Features | Preprocessing Steps | Data-Splitting Method |
|---|---|---|---|---|---|---|---|---|---|
| Z. Chen et al., 2017 [35] | No | N/A | N/A | N/A | N/A | N/A | N/A | Students' score categorization (A, B, C). | N/A |
| Z. Chen et al., 2020 [36] | No | 18,457 | 2022 | 309 | N/A | N/A | N/A | N/A | K-fold cross-validation. |
| Z. Ren et al., 2019 [37] | No | N/A | 43,099 | N/A | 151 | N/A | N/A | Used different embedding dimensions for students, courses, and course instructors for different majors. | Nested time-series split cross-validation (data from Fall 2009 to Fall 2015 as a training set, and data from Spring 2016 as a testing set). |

6.2.1. Dataset Description of Studies That Used Collaborative Filtering

Table 10 shows the dataset description of each study that used the collaborative filtering algorithm in their proposed recommender system. Additionally, any preprocessing steps mentioned in the research, as well as the data-splitting method.

The datasets used in 12 out of 13 collaborative filtering research papers were private. Most collaborative filtering research papers did not provide information about the used dataset.

Additionally, five research papers have not mentioned any information about any preprocessing steps performed on the used dataset, and only one research paper mentioned the number of used features.

Moreover, three research papers used K-fold cross-validation for data splitting, however, only two papers mentioned the value of k. Seven research papers did not provide any information about the data-splitting method that was used.

6.2.2. Dataset Description of Studies That Used Content-Based Filtering

Table 11 summarizes the dataset information found in the two content-based filtering studies that were included in the survey. The two content-based filtering studies used a private dataset. Especially when utilizing private datasets, researchers ought to provide detailed information about the used dataset and the splitting method.

One research paper [38] provided relatively detailed information about the used private dataset, as well as the data-splitting method and percentages of training and testing sizes. On the other hand, the other research [39] has not provided any information about the used dataset or the data-splitting method. However, the two research papers provided detailed information on the preprocessing steps performed.

**Table 11.** Description of datasets used in each content-based filtering study, the preprocessing steps, and the data-splitting method.

| Authors and Year | Public | Records | Students | Courses | Majors | Features | Features Used | Preprocessing Steps | Data-Splitting Method |
|---|---|---|---|---|---|---|---|---|---|
| A. J. Fernández-García et al., 2020 [38] | No | 6948 | 323 | N/A | N/A | 10 | 10 | • Feature deletion, • Class reduction, • One-hot encoding, • Creating new features, • Data scaling: MinMax Scaler, Standard Scaler, Robust Scaler, and Normalizer Scaler, • Data resampling: upsample, downsample, SMOTE. | • Train size = 80%, • Test size = 20%. |
| Y. Adilaksa et al., 2021 [39] | No | N/A | N/A | N/A | N/A | N/A | N/A | • Case folding, • Word tokenization, • Punctuation removal, • Stop words removal. | N/A |

### 6.2.3. Dataset Description of Studies That Used Hybrid Recommender Systems

Table 12 illustrates the dataset description and information for the six studies that used the hybrid recommender system approach for filtering. It is noticeable that most research papers utilizing this approach used datasets with a very small number of students.

**Table 12.** Description of datasets used in each hybrid recommender system study, the preprocessing steps, and the data-splitting method.

| Authors and Year | Pub | Recs | Students | Courses | Majors | Features | Used Features | Preprocessing Steps | Data-Splitting Method |
|---|---|---|---|---|---|---|---|---|---|
| Esteban, A. et al., 2020 [40] | No | 2500C | 95 | 63 | N/A | N/A | N/A | N/A | Five-fold cross-validation. |
| M. I. Emon et al., 2021 [41] | No | N/A | 250+ | 250+ | 20+ | N/A | N/A | Feature extraction. | N/A |
| S. Alghamdi et al., 2022 [42] | No | 1820 | 38 | 48 | N/A | N/A | 7 | Cluster sets for academic transcript datasets. | Five-fold cross-validation. |
| S. G. G et al., 2021 [43] | No | N/A | ~6000 | ~4000 | 18 | N/A | N/A | N/A | N/A |
| S. M. Nafea et al., 2019 [44] | No | N/A | 80 | N/A | N/A | N/A | N/A | N/A | Student dataset was split into cold-start students, cold-start learning objects, and all students. |
| X. Huang et al., 2018 [45] | Yes | N/A | 56,600 | 860 | N/A | N/A | N/A | N/A | • Train size = 80%, • Test size = 20%. |

Five research papers used private datasets and only [45] used a public dataset. Most research studies utilizing this approach have not provided any information on any pre-processing steps performed on the dataset. Additionally, authors in [41,43] provided

approximate numbers for the count of students and courses in the dataset, they did not explain why the exact numbers were not provided.

### 6.2.4. Dataset Description of Studies That Used Novel Approaches

Table 13 shows the dataset description and information for the 13 research papers that presented various novel approaches for recommender systems. Ten research papers out of thirteen provided information about the performed preprocessing steps. Additionally, the same number of papers provided information about the data-splitting method. The main three data-splitting methods were utilized in research presenting these novel approaches:

- Train-test split.
- K-fold cross-validation.
- Nested time series splits.

**Table 13.** Description of datasets used in each novel approach recommender system study, the preprocessing steps, and the data-splitting method.

| Authors and Year | Public | Records | Students | Courses | Majors | Features | Features Used | Preprocessing Steps | Data-Splitting Method |
|---|---|---|---|---|---|---|---|---|---|
| A. Baskota et al., 2018 [46] | No | 16,000 | N/A | N/A | N/A | N/A | N/A | ● Data cleaning, ● Data scaling. | ● Train size = 14,000, ● Test size = 2000. |
| Jiang, Weijie et al., 2019 [47] | No | 4,800,000 | 164,196 | 10,430 | 17 | N/A | N/A | N/A | Nested time-series split cross-validation (data from F'08 to F'15 as a training set, data in Sp'16 as validation set & data in Sp'17 as test set) |
| Liang, Yu et al., 2019 [48] | No | 35,000 | N/A | N/A | N/A | N/A | N/A | Data cleaning. | N/A |
| M. Isma'il et al., 2020 [49] | No | 8700 | N/A | 9 | N/A | N/A | 4 | ● Data cleaning, ● Data encoding. | N/A |
| M. Revathy et al., 2022 [50] | No | N/A | 1243 | N/A | N/A | N/A | 33 | ● One-hot encoding for categorical features, ● Principal Component Analysis (PCA). | ● Train size = 804, ● Test size = 359. |
| Oreshin et al., 2020 [51] | No | N/A | >20,000 | N/A | N/A | N/A | 112 | ● One-hot encoding, ● Removed samples with unknown values. | Nested time-series split cross-validation. |
| R. Verma et al., 2018 [52] | No | 658 | 658 | N/A | N/A | 13 | 11 | Data categorization. | Ten-fold cross-validation. |

**Table 13.** *Cont.*

| Authors and Year | Public | Records | Students | Courses | Majors | Features | Features Used | Preprocessing Steps | Data-Splitting Method |
|---|---|---|---|---|---|---|---|---|---|
| S. D. A. Bujang et al., 2021 [53] | No | 1282 | 641 | 2 | N/A | 13 | N/A | ● Ranked and grouped the students into five categories of grades, ● Applied oversampling SMOTE (Synthetic Minority Over-sampling Technique), ● Applied two feature selection methods: Wrapper and filter-based. | Ten-fold cross-validation. |
| S. Srivastava et al., 2018 [54] | No | 1988 | 2890 | N/A | N/A | N/A | 14 | Registration number transformation. | ● Train = 1312, ● Test = 676. |
| T. Abed et al., 2020 [55] | No | N/A | N/A | N/A | N/A | N/A | 18 | Balanced the dataset using under sampling. | Ten-fold cross-validation. |
| V. L. Uskov et al., 2019 [56] | No | 90+ | N/A | N/A | N/A | 16 | N/A | Data cleaning | ● Train = 80%, ● Test = 20%. |
| V. Sankhe et al., 2020 [57] | No | N/A | 2000 | 15 | 7 | N/A | N/A | N/A | N/A |
| V. Z. Kamila et al., 2019 [58] | No | N/A | N/A | N/A | N/A | N/A | N/A | N/A | ● Train size = 75%, ● Test size= 25%. |

### 6.2.5. Dataset Description of the Study That Used Similarity-Based Filtering

Table 14 shows the dataset description of the only research paper that used the similarity-based filtering recommender system. Two private datasets were utilized in this research with relatively small sizes. The authors have not provided any information about the preprocessing steps performed on the datasets in the research. It is also noticed that the authors used a relatively small test set.

**Table 14.** Description of the dataset used the only similarity-based filtering recommender system study, the preprocessing steps, and the data-splitting method.

| Authors and Year | Public | Records | Students | Courses | Majors | Features | Features Used | Preprocessing Steps | Data-Splitting Method |
|---|---|---|---|---|---|---|---|---|---|
| D. Shah et al., 2017 [59] | No | N/A | ● Dataset 1 = 300 students ● Dataset 2 = 84 students | ● Dataset 1 = 10 ● Dataset 2 = 26 | N/A | N/A | ● Student features = 3 ● Course features = 30 | N/A | ● Train size = 90% ● Test size = 10% |

### 6.3. Research Evaluation

This subsection summarizes the weakness and strength points of each research paper included in the survey, as well as the type of performance evaluation metrics used and their corresponding values.

6.3.1. Research Evaluation for Studies That Used Collaborative Filtering

Table 15 shows the evaluation metrics of each collaborative filtering study and their values. Additionally, all the observed strengths and weaknesses are listed. The used evaluation metrics ranged from very well-known, such as Mean Absolute Error (MAE) and Root Mean Square Error (RMSE), to very uncommon metrics, such as AverHitRate and AverAcc. This is also the case with all other approaches.

**Table 15.** The performance metrics used, the performance results, the strengths, and the weaknesses points of studies that used collaborative filtering.

| Authors and Year | Evaluation Metrics and Values | Strengths | Weaknesses |
|---|---|---|---|
| A. Bozyiğit et al., 2018 [26] | MAE = 0.063. | • Compared the performance of the proposed OWA approach with the performance of other popular approaches. | • The number of features and features used in the dataset is not provided,<br>• The dataset description is not detailed,<br>• Did not use RMSE for evaluation, considered the standard as it's more accurate,<br>• Mentioned that some preprocessing had been carried out but did not give any details regarding it. |
| B. Mondal et al., 2020 [27] | • MSE = 3.609,<br>• MAE = 1.133,<br>• RMSE = 1.8998089,<br>• Precision,<br>• Recall. | • Used many metrics for evaluation,<br>• The implementation of algorithms is comprehensively explained. | • Did not mention whether they split data for testing or used the training data for testing,<br>• Did not provide the exact measures of precision and recall. |
| E. L. Lee et al., 2017 [28] | AUC = 0.9709. | • Compared the performance of the proposed approach with the performance of other approaches,<br>• Used a very large dataset,<br>• Achieved a very high AUC,<br>• The implementation of algorithms is comprehensively explained. | • Did not provide the percentage of the train-test split,<br>• The number of courses in the dataset is not mentioned (it only mentions course registration records). |
| I. Malhotra et al., 2022 [29] | • MAE = 0.468,<br>• RMSE = 0.781. | • The implementation of algorithms is comprehensively explained with examples,<br>• Used RMSE and MAE for evaluation. | • The dataset description is not detailed,<br>• The method of splitting the training and testing dataset is not provided,<br>• Did not mention whether they have done any preprocessing on the dataset or if it was used as it is,<br>• The proposed approach is not compared to any other approaches in the evaluation section. |
| L. Huang et al., 2019 [30] | • AverHitRate between 0.6538, 1,<br>• AverACC between 0.8347, 1. | • The literature is meticulously discussed,<br>• The implementation is comprehensively explained in detail. | • The method of splitting the training and testing dataset is not provided,<br>• Did not mention whether they have conducted any preprocessing on the dataset or if it was used as it is. |

**Table 15.** *Cont.*

| Authors and Year | Evaluation Metrics and Values | Strengths | Weaknesses |
|---|---|---|---|
| L. Zhao et al., 2021 [31] | • Precision, <br> • Recall. | • The implementation is comprehensively explained. | • The exact numbers for the evaluation metrics used in the paper are not provided, <br> • The numbers of features and features used in the dataset are not provided. |
| M. Ceyhan et al., 2021 [32] | • Coverage, <br> • F1-measure, <br> • Precision, <br> • Sensitivity, <br> • Specificity, <br> • MAE, <br> • RMSE, <br> • Binary MAE, <br> • Binary RMSE. | • Used many metrics for evaluation. | • The implemented algorithm and similarities explanation were very brief |
| R. Obeidat et al., 2019 [9] | • Coverage measure (using SPADES ｜ with clustering) = 0.376, 0.28, 0.594, 0.546, <br> • Coverage measure (using Apriori ｜ with clustering) = 0.46, 0.348, 0.582, 0.534. | • Confirmed by experiment that clustering significantly improves the generation and coverage of two association rules: SPADES and Apriori | • The dataset description is not detailed, <br> • The method of splitting the training and testing dataset is not provided, <br> • The implementation is not discussed in detail. |
| S. Dwivedi et al., 2017 [33] | • RMSE = 0.46. | • The proposed system is efficient as it proved to work well with big data, <br> • The implementation of algorithms is comprehensively explained. | • Did not provide any information about the dataset, <br> • The literature review section was very brief. |
| S.-T. Zhong et al., 2019 [34] | • MAE (CS major) = 6.6764 ± 0.0029, <br> • RMSE (CS major) = 4.5320 ± 0.0022. | • Used eight datasets for model training and evaluation, <br> • Dataset description is detailed, <br> • Compared the performance of the proposed approach with the performance of other popular approaches. | • The percentage of train-test splitting is not consistent among the eight datasets. |
| Z. Chen et al., 2017 [35] | • Confidence, <br> • Support. | • The implementation of algorithms is comprehensively explained with examples. | • Did not provide any information about the used dataset, <br> • Did not include any information about the preprocessing of the dataset, <br> • Did not provide useful metrics for evaluation, <br> • The performance of the proposed approach is not compared to other similar approaches. |

**Table 15.** *Cont.*

| Authors and Year | Evaluation Metrics and Values | Strengths | Weaknesses |
|---|---|---|---|
| Z. Chen et al., 2020 [36] | • Precision,<br>• Recall,<br>• F1-score. | • Compared the performance of the proposed approach with the performance of other popular approaches: cosine similarity and improved cosine similarity. | • The exact numbers for the evaluation metrics used in the paper are not provided,<br>• The numbers of features and features used in the dataset are not provided. |
| Z. Ren et al., 2019 [37] | • PTA,<br>• MAE. | • Compared the performance of the proposed approach with the performance of other approaches,<br>• The implementation of algorithms is comprehensively explained,<br>• The number of students in the dataset is big. | • The dataset description is not detailed. |

**Key**: MAE is Mean Absolute Error; OWA is Ordered Weighted Averaging; RMSE is Root Mean Squared Error; MSE is Mean Squared Error; AUC is Area Under the Curve; AverHitRate is Average Hit Rate; AverACC is Average Accuracy; SPADES is a type of algorithm used for mining frequent patterns in the given data; Apriori is a popular algorithm for mining frequent item sets for Boolean association rules; PTA is Percentage of Tick Accuracy; CS refers to Computer Science in this context.

### 6.3.2. Research Evaluation for Studies That Used Content-Based Filtering

Table 16 shows the evaluation metrics used in the two content-based filtering studies along with their values. Moreover, the weaknesses and strength points observed in the two papers are listed. This research area contained a research paper [38]. Where no weaknesses could be addressed.

**Table 16.** The performance metrics used, the performance results, the strength points, and the weaknesses of studies that used content-based filtering.

| Authors and Year | Evaluation Metrics and Values | Strengths | Weaknesses |
|---|---|---|---|
| A. J. Fernández-García et al., 2020 [38] | • Accuracy,<br>• Precision,<br>• Recall,<br>• F1-score. | • Included a section that contains the implementation code,<br>• The literature is meticulously discussed and followed by a table for a summary,<br>• Compared the effect of various preprocessing steps on the final measures of different machine-learning approaches and provided full details about these metrics,<br>• The implementation of each preprocessing step is explained in detail. | • N/A |
| Y. Adilaksa et al., 2021 [39] | • The percentage of recommendation diversity = 81.67%,<br>• Accuracy = 64%. | • The preprocessing steps are discussed in detail,<br>• The implementation is comprehensively explained,<br>• Confirmed by the experiment that using the weighted cosine similarity instead of the traditional cosine similarity significantly increased the accuracy of the course recommendations system. | • Did not provide any information about the used dataset,<br>• The method of splitting the training and testing dataset is not provided,<br>• The accuracy measurement is not specified. |

### 6.3.3. Research Evaluation for Studies That Used Hybrid Recommender Systems

Table 17 shows the evaluation metrics for hybrid recommender system studies along with their observed weaknesses and strength points. It is noticed that the most common



weakness among all papers is the ambiguity of the used dataset because of the lack of provided information combined with the use of a private dataset.

**Table 17.** The performance metrics used, the performance results, the strength points, and the weaknesses of studies that used hybrid recommender systems.

| Authors and Year | Evaluation Metrics and Values | Strengths | Weaknesses |
|---|---|---|---|
| Esteban, A. et al., 2020 [40] | • RMSE = 0.971, <br> • Normalized discount cumulative gain (nDCG) = 0.682, <br> • Reach = 100%, <br> • Time = 3.022s. | • The literature is meticulously discussed and followed by a table for a summary, <br> • The implementation of algorithms is comprehensively explained with examples, <br> • Compared the performance of the proposed hybrid approach with other similar approaches, <br> • Used many useful metrics for evaluation. | • Mentioned that some preprocessing had been carried out but did not give any details regarding it, <br> • The number of students in the dataset is relatively low. |
| M. I. Emon et al., 2021 [41] | • Accuracy, <br> • Precision, <br> • Recall, <br> • F1-score. | • Compared the performance of the proposed hybrid approach with the used standalone algorithms. | • The exact numbers for the evaluation metrics used in the paper are not provided, <br> • The dataset description is not detailed, <br> • The method of splitting the training and testing dataset is not provided. |
| S. Alghamdi et al., 2022 [42] | • MAE = 0.772, <br> • RMSE = 1.215. | • The dataset description is detailed, <br> • The implementation of algorithms is clearly explained. | • Other similar approaches are not stated in the literature, <br> • The number of students in the dataset is relatively low. |
| S. G. G et al., 2021 [43] | RMSE = 0.931. | • EDA of the dataset is included in the paper, <br> • Compared the performance of different approaches against the proposed approach, <br> • The implementation is comprehensively discussed and explained. | • The dataset description is not detailed, <br> • The method of splitting the training and testing dataset is not provided, <br> • Similar approaches are not stated in the literature, <br> • Did not mention whether they conducted any preprocessing on the dataset or if it was used as it is. |
| S. M. Nafea et al., 2019 [44] | • MAE for cold students = 0.162, <br> • RMSE for cold students = 0.26, <br> • MAE for cold Learning Objects (Los) = 0.162, <br> • RMSE for cold LOs = 0.3. | • Achieved higher accuracy than standalone traditional approaches mentioned in the paper: collaborative filtering and content-based recommendations, <br> • The implementation is comprehensively explained with examples. | • Mentioned that some preprocessing had been carried out but did not give any details regarding it, <br> • The dataset description is not detailed, <br> • The number of students in the dataset is relatively low. |
| X. Huang et al., 2018 [45] | • Precision, <br> • Recall, <br> • F1-score. | • The implementation of the proposed approach is comprehensively explained with examples, <br> • Compared the performance of the proposed hybrid approach with other similar approaches through testing. | • The dataset description is not detailed, <br> • Did not mention whether they have done any preprocessing on the dataset or if it was used as it is, <br> • The exact numbers for the evaluation metrics used in the paper are not provided. |

6.3.4. Research Evaluation for Studies That Used Novel Approaches

Table 18 shows the performance metrics for the studies that used novel approaches along with the weaknesses and strength points of each one. This research area contains the second research [50], where no weaknesses could be addressed.

**Table 18.** The performance metrics used, the performance results, the strength points, and the weaknesses of studies that used novel approaches.

| Authors and Year | Evaluation Metrics and Values | Strengths | Weaknesses |
|---|---|---|---|
| A. Baskota et al., 2018 [46] | • Accuracy = 61.6%, <br> • Precision = 61.2%, <br> • Recall = 62.6%, <br> • F1-score = 61.5%. | • Compared the performance of the proposed approach with the performance of other popular approaches, <br> • Used many evaluation metrics and provided the exact numbers for each metric for the evaluation result. | • The dataset description is not detailed. |
| Jiang, Weijie et al., 2019 [47] | • The A model: accuracy = 75.23%, F-score = 60.24%, <br> • The B model: accuracy = 88.05%, F-score = 42.01%. | • The implementation of algorithms is comprehensively explained with examples, <br> • Included various sets of hyperparameters and carried out extensive testing. | • Did not mention whether they have done any preprocessing on the dataset or if it was used as it is, <br> • Did not mention the number of features in the dataset, <br> • The performance of the proposed approach is not compared to other similar approaches, <br> • Did not mention the exact percentages for splitting data. |
| Liang, Yu et al., 2019 [48] | • Support rate. | • The implementation of algorithms is comprehensively explained. | • The dataset description is not detailed, <br> • A literature review has not been discussed, <br> • The performance of the proposed approach is not compared to other similar approaches, <br> • Did not provide many useful metrics for evaluation and explained that was due to the large number of data sets selected for the experiment. |
| M. Isma'il et al., 2020 [49] | • Accuracy = 99.94%. | • Compared the performance of the proposed machine-learning algorithm with the performance of other algorithms through testing. | • Did not mention the training and test set sizes, <br> • The machine learning algorithms used are not explained, <br> • Only used the accuracy measure for evaluation, <br> • The dataset description is not detailed. |
| M. Revathy et al., 2022 [50] | • Accuracy = 97.59%, <br> • Precision = 97.52%, <br> • Recall = 98.74%, <br> • Sensitivity = 98.74%, <br> • Specificity = 95.56%. | • Used many evaluation metrics and provided the exact numbers for each metric for the evaluation result, <br> • Provided detailed information about the preprocessing steps, <br> • Compared the performance of the proposed approach with the performance of other approaches, <br> • Provided the exact numbers for each metric for the evaluation result. | N/A |

<div align="center">

**Table 18.** *Cont.*

</div>

| Authors and Year | Evaluation Metrics and Values | Strengths | Weaknesses |
|---|---|---|---|
| Oreshin et al., 2020 [51] | • Accuracy = 0.91 ± 0.02, <br> • ROC-AUC = 0.97 ± 0.01, <br> • Recall = 0.83 ± 0.02, <br> • Precision = 0.86 ± 0.03. | • Used many evaluation metrics and provided the exact numbers for each metric for the evaluation result, <br> • Provided detailed information about the preprocessing steps. | • Contains many English grammar and vocabulary errors, <br> • The dataset description is not detailed, <br> • The machine learning algorithms used are not explained, <br> • Did not specify the parameters for the nested time-series split cross-validation. |
| R. Verma et al., 2018 [52] | • Accuracy (SVM) = 88.5%, <br> • Precision, <br> • Recall, <br> • F1-score. | • The implementation of algorithms is comprehensively explained, <br> • Compared the performance of several machine-learning algorithms with the performance of other algorithms through testing and concluded that the best two were SVM and ANN. | • The exact numbers for the evaluation metrics used in the paper are not provided except for the achieved accuracy of SVM. |
| S. D. A. Bujang et al., 2021 [53] | • Accuracy = 99.5%, <br> • Precision 99.5%, <br> • Recall = 99.5%, <br> • F1-score = 99.5%. | • Included all the exact numbers for the evaluation metrics used in the evaluation, <br> • Compared the performance of six machine learning algorithms and concluded that random forests performed the best based on the evaluation metrics, <br> • EDA of the dataset is included in the paper, <br> • The literature is meticulously discussed and followed by a table for a summary, <br> • Provided detailed information about the used dataset. | • The number of courses is very low (only 2). |
| S. Srivastava et al., 2018 [54] | • Accuracy (from 1 cluster to 100) = 99.40%:87.72%. | • Compared the performance of the proposed approach with the performance of other popular approaches, <br> • Provided a confusion matrix for all the used approaches. | • Accuracy is the only metric used for evaluation, <br> • The dataset description is not detailed. |
| T. Abed et al., 2020 [55] | • Accuracy = 69.18%. | • Compared the performance of the proposed approach with the performance of other popular approaches: Random Forest, J48, Naive Bayes, Logistic Regression, Sequential Minimal Optimization, and a Multilayer Perceptron. | • The dataset description is not detailed, <br> • Only used the accuracy measure for evaluation, <br> • Did not include an explanation for the implemented algorithms and why they were initially chosen. |
| V. L. Uskov et al., 2019 [56] | • Average error = 3.70%. | • Through extensive testing of various ML algorithms, they concluded that linear regression was the best candidate for the problem as the data was linear; <br> • The implementation of algorithms is comprehensively explained. | • The dataset description is not detailed, <br> • Only used the accuracy measure for evaluation, <br> • Did not use RMSE for the evaluation of linear regression. |

**Table 18.** *Cont.*

| Authors and Year | Evaluation Metrics and Values | Strengths | Weaknesses |
|---|---|---|---|
| V. Sankhe et al., 2020 [57] | • Accuracy = 81.3% | • The implementation of algorithms is comprehensively explained. | • The dataset description is not detailed, • The method of splitting the training and testing dataset is not provided, • Did not mention whether they have conducted any preprocessing on the dataset or if it was used as it is. |
| V. Z. Kamila et al., 2019 [58] | • Accuracy of KNN K = 1:100.00% • Accuracy of Naive Bayes algorithm = 100.00% | • Provided the exact numbers for each metric for the evaluation result. | • The implemented algorithms explanation was very brief, • The performance of the proposed approach is not compared to other similar approaches, • Did not provide any information about the dataset used, • Did not mention whether they have conducted any preprocessing on the dataset or if it was used as it is. |

Again, the most common weaknesses in this research area were dataset-related, whether the authors did not mention any information about the used dataset, as in [58]; or the ambiguity about preprocessing steps, performed as [47,57,58]; or the lack of information about the data-splitting methods used, as in [48,49,57].

6.3.5. Research Evaluation for the Study That Used Similarity-Based Filtering

Table 19 shows the performance evaluation metrics for the only research that used similarity-based filtering that is included in this literature survey. Additionally, all the weaknesses and strength points observed in the mentioned study are listed.

**Table 19.** The performance metrics used, the performance results, the strength points, and the weaknesses of the only study that used similarity-based filtering.

| Authors and Year | Evaluation Metrics | Strengths | Weaknesses |
|---|---|---|---|
| D. Shah et al., 2017 [59] | • Normalized mean absolute error (NMAE) = 0.0023, • Computational Time Comparison. | • The implementation of the two compared algorithms is comprehensively explained, • Compared the accuracy of recommendations from both algorithms as well as the speed. | • Did not mention whether they have conducted any preprocessing on the dataset or if it was used as it is, • Similar approaches are not stated in the literature, in addition, the literature was very brief, • Did not use RMSE for evaluation, which is considered the standard as its more accurate. |

This research paper used an evaluation metric that is very important but was not utilized by any other research which is the computational time comparison. This metric is important as the recommendations need to be filtered quickly.

### 7. Discussion of Findings

Figure 4 shows the total number of different recommender system implementations that are included in the review. It shows that (i) collaborative filtering-based approaches, and (ii) novel approaches represent most of the research papers discussed in the literature.

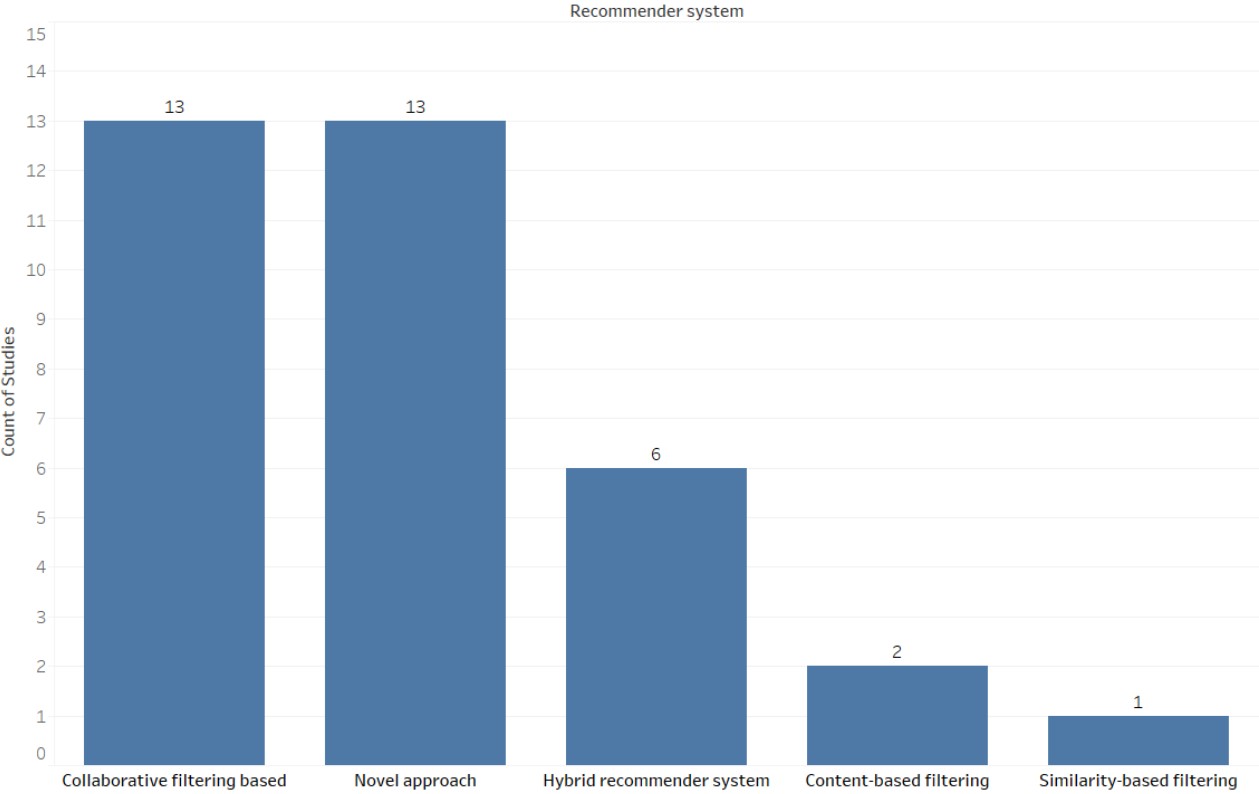

**Figure 4.** Type of recommender system publication count.

There are a total of 35 research papers discussed in this SLR. A total of thirteen papers use collaborative filtering-based approaches, thirteen use novel approaches, six use the hybrid recommender systems approach, two use the content-based filtering approach, and only one research paper uses the similarity-based filtering approach. The selection of a recommender system algorithm in these papers is influenced by several variables, including the available data, the particular advising system objectives, and the environment in which it is being utilized.

The foundation of collaborative filtering is the notion that individuals will continue to have similar preferences as they had in the past. This suggests that students who have taken related courses and fared well in them are likely to have related interests and perform well in subsequent related courses. The usage of user-based or item-based filtering is only one example of the several methods that collaborative filtering algorithms might employ to find commonalities among students.

On the other hand, content-based filtering suggests courses based on the course characteristics, such as course descriptions, syllabi, prerequisites, and learning outcomes. This kind of screening aligns the course features with the choices and interests of the learner.

Academic advising systems are also utilizing hybrid systems more frequently since they may deliver suggestions that are more individualized and precise by combining collaborative and content-based filtering. Although collaborative filtering, content-based filtering, and hybrid recommender systems are frequently employed in academic advising systems to recommend courses, each of these techniques has significant problems of their own.

Collaborative filtering has the problem known as a "cold start", which occurs when a new student or course is added to the system. Collaborative filtering may have trouble providing precise recommendations for new users or products because it is dependent on the users' prior behavior. Additionally, to identify similar students and courses, collaborative filtering needs a substantial amount of student–course interaction data. Sometimes, there might not be enough information, which would cause recommendations to be less precise. Finally, collaborative filtering is frequently seen as a "black box," which makes it difficult to understand how the algorithm came up with a certain recommendation. Students may find it difficult to comprehend why specific courses were suggested to them as a result. Finally, the recommendations may not be right if the data defining the course features are inaccurate or incomplete.

Content-based filtering has the problem of diversity limitation because content-based filtering depends on a course's features, and it could be difficult to suggest courses that go beyond a student's previous coursework or areas of interest. Additionally, content-based filtering systems usually suffer from over-specialization if the system depends too much on particular features, as it can suggest classes that are too close to ones the student has already taken, which would limit their overall academic experience.

Hybrid recommender systems might be more difficult and expensive to install since they demand more resources to create and maintain. Moreover, merging several recommendation algorithms might be difficult; this calls for thorough calibration and integration to guarantee that the system produces correct recommendations. The system may overfit the data if it is too complicated, which will result in less precise recommendations.

Ultimately, each approach has its advantages and disadvantages, and the best course of action will rely on the objectives and limitations of the advising system. To make sure the system gives students accurate and helpful recommendations, it is important to carefully weigh these elements and assess the system's effectiveness.

Notably, we have witnessed a significant increase in recommender systems that avoided using the previous traditional approaches and instead presented novel approaches for recommender systems that seem to help against the shortcomings of the traditional approaches. This spike in using novel approaches for course recommendation systems happened between 2018 and 2020, and since then, the number of research papers presenting novel approaches decreased significantly. Although, from the literature, we can conclude that some novel approaches provided both comprehensive and precise recommendations. Nevertheless, the provided results are dependent on many variables such as the size of the dataset, and the number of features used.

In the previous two years, the research in CRSs was heading towards utilizing traditional approaches such as collaborative filtering recommender systems and hybrid recommender systems, and the presentation of novel approaches was noticeably decreasing, although it is probably the most promising as the industry is heading away from traditional approaches.

Figure 5 shows that most recommender system publications in the area of course recommendations were on the algorithm level, while two publications were on the preprocessing-level type of comparative, and only one publication was on both the algorithm and preprocessing levels [53].

Most studies did not provide enough information about the dataset that was used. Furthermore, some studies used very small datasets. This is considered a major weakness because recommender system performance metrics can be misleading. Moreover, many studies mentioned that some preprocessing had been carried out and did not provide any information about it. Instead, some publications did not mention any information about whether preprocessing had been carried out or not. Data preprocessing is a vital phase in any recommender system design, so it is important to discuss it.

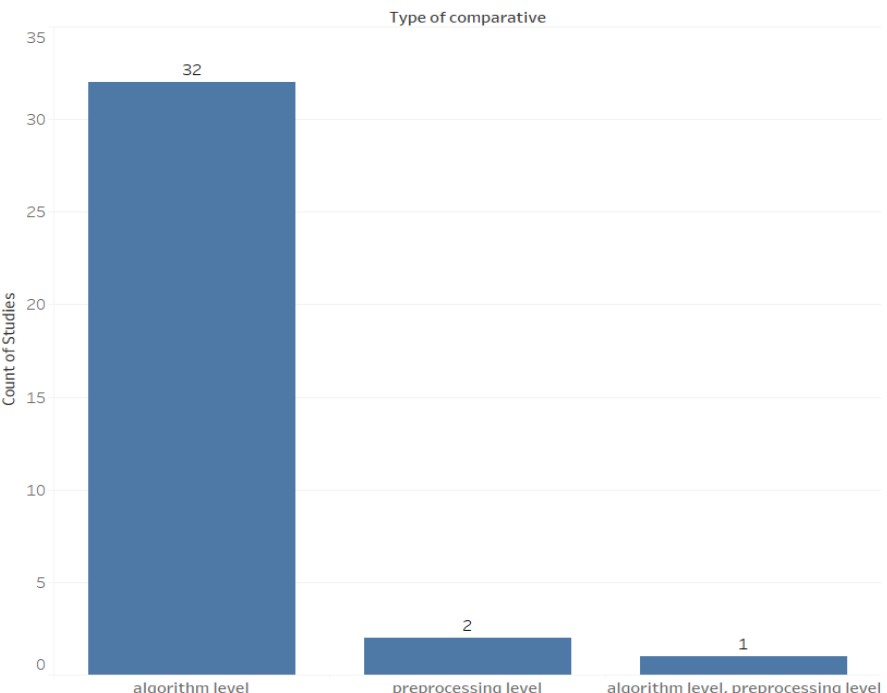

**Figure 5.** Count of comparative types of publications.

Additionally, many publications have not mentioned the data-splitting method used in the implementation or the splitting ratio. Many data-splitting methods were used, including train-test splitting, K-fold splitting, and time series splitting. Arguably, the best splitting method is K-fold because it makes use of all the data in both training and testing. However, K-fold is the most computationally expensive method. True because from each training dataset, numerous models will be created, and on each test dataset, those models will be tested. While this is not a drawback for small datasets, when the models are vast and the datasets are enormous, things soon become very expensive.

Many of the studies exhibited common weaknesses and/or strengths, with the most frequent weakness being little or no information provided about the dataset. Many papers failed to provide information about the data-splitting method. Furthermore, many studies mentioned the use of some evaluation metrics and did not include the exact performance results for some and/or any of these metrics. On the other hand, some papers, such as [32,38], included arguably, too many metrics for evaluation on different datasets. Finally, some publications did not discuss the implementation of the proposed algorithm in enough detail to allow reproduction of the results.

Finally, most research papers have not compared the performance of the proposed system to the performance of other models from the various publications in the targeted research area. The weakness, however, is probably because most datasets used in the implementation of these algorithms are not public datasets (i.e., only 3 studies utilized public datasets).This is a noticeable obstacle in the development of research in CRSs.

## 8. Gaps, Challenges, Future Directions and Conclusions for (CRS) Selection

Society's decisions in a variety of areas, including entertainment; shopping; and, increasingly, education, are influenced by recommender systems, which have become an essential component of our digital life. For navigating the complicated world of academic options in higher education, course selection recommender systems specifically offer a promising tool. To help students make educated judgments about their academic routes, these platforms use machine-learning algorithms to deliver individualized course recommendations.

This technology, however, faces several difficulties and has potential for development as it continues to advance. The intricacy of these concerns is explored in-depth in this chapter, from the technical challenges of maintaining enormous and diverse data to the moral ramifications of justice, bias, and privacy.

We will examine the necessity of multidisciplinary cooperation in addressing these issues in greater detail, highlighting how a fusion of computer science, education, psychology, and data science can result in more dependable and robust systems. We will also talk about the practical consequences and prospective uses of cutting-edge CRSs in educational institutions.

This section's goal (i.e., research gaps, challenges, and future directions) is to provide a thorough review of the state of CRSs (Course Recommender Systems) today, their problems, and open questions for further study and development. This section can act as a reference point for academics, teachers, and public officials interested in the relationship between artificial intelligence and education.

### 8.1. Gap

One of the noticeable gaps is the lack of substantial details and validation regarding the datasets used in the studies. Many studies failed to disclose crucial information about the datasets, such as their size, characteristics, and sources. As a result, it is challenging to compare and evaluate the performance of different recommender systems objectively. Future studies should put more effort into transparency about their datasets especially. Otherwise, consider using public datasets to enable more direct comparisons and evaluations.

Preprocessing presents yet another important omission. Preprocessing is a key step in any recommender system, yet it is frequently ignored or just briefly covered within the literature we reviewed. It is difficult to comprehend the precise actions taken to prepare the data for recommendation generation because of this lack of specificity. To fully comprehend the effects of various preprocessing methods on the effectiveness of various recommender systems for course selection, more effort in transparency is required.

Finally, there is a notable lack of research that finally assesses the effectiveness of the CRS algorithms from the same dataset. Most studies tend to concentrate on one or a small number of algorithms without comparing their performance to that of other well-known algorithms. This causes a gap in our knowledge of how various algorithms compare to one another when used for course selection.

### 8.2. Challenges

The "cold start" problem is a significant challenge in recommender systems for course selection, especially in collaborative filtering-based systems. When a new student or a new course is added, these systems struggle to provide accurate recommendations due to the lack of prior contextual behavior or data. This issue necessitates the development of innovative strategies, techniques or algorithms that can deal with this problem effectively.

Another difficulty is diversity limitation, especially for content-based filtering algorithms. These systems might have a hard time suggesting classes for students that are not related to their prior coursework or areas of interest, which might limit their academic experience. To solve this problem, new approaches or procedures must be created that can increase the variety of suggestions.

Lastly, it can be difficult to develop hybrid recommender systems because of their complexity and cost. To ensure correct suggestions, these systems need greater resources to build and maintain, as well as thorough calibration and integration. To develop ways to lower the complexity and cost of these systems without sacrificing their performance, more research is required.

### 8.3. Future Directions

Future studies for comparing the efficacy of CRSs for choosing to recommend certain courses might concentrate on several different topics. The creation of creative solutions to

the cold-start issue in collaborative filtering-based systems is one such topic. This can entail looking for ways to incorporate auxiliary information, such as demographic data, into the recommendation process. For example, what are the stopping criteria and to efficiently verify that recommendations are feasible and correct.

Another promising area for future research is the development of methods for enhancing the diversity of recommendations in content-based filtering systems. This could involve exploring techniques for broadening the range of course features considered during the recommendation process, or techniques for integrating collaborative filtering elements into content-based systems to leverage the benefits of both approaches.

Future studies may find success using sophisticated machine-learning and deep-learning methods. These methods might improve the memory and accuracy of recommendations, especially when combined with other recommendation techniques.

Future studies could leverage well known data engineering methods to determine how other data sources, such as social media and Internet-browsing history, could enhance the accuracy and personalization of course recommendations. These data sources might offer insightful information on the interests, preferences and predelections of students, which may improve the caliber of recommendations.

While there are significant challenges and gaps in the current research on recommender systems for course selection, there are also many promising avenues for future work. By addressing these challenges and exploring these future directions, we can aim to develop more robust, accurate, and personalized recommender systems that truly meet the individual needs of students (i.e., at least classes of student learners).

Advancements in machine learning and artificial intelligence provide the potential to significantly enhance the predictive accuracy, precision and relevancy of these systems. For instance, incorporating deep-learning algorithms can allow the systems to discern complex patterns and dependencies in student data, thereby improving the quality of recommendations. Moreover, the use of reinforcement learning could help in continuously adapting the system based on the feedback from students, enabling a more dynamic and personalized course recommendation process.

In addition to technical improvements, there is a need for larger, more varied datasets that reflect a variety of student characteristics, learning preferences, and academic backgrounds. The ability of recommender systems to generalize and provide precise recommendations for a larger range of pupils depends on how diverse and extensive the data is. Additionally, the development of standardized datasets that are open to the public may help researchers in the field share best practices and enable more comparitive studies.

The fusion of recommender systems with other educational technology is a further potential direction. A more comprehensive understanding of a student's performance and preferences, for instance, could be obtained by integrating these systems with learning-management systems, allowing for more precise course recommendations. Integration with career guidance programs could also improve matching course recommendations with the student's career goals and aspirations. This would not only create an individualized learning pathway but also help students envision their academic journey's potential professional outcomes.

Future research should address the concerns of fairness, bias, and privacy in recommender systems on the ethical and social fronts. It is crucial to ensure that the processes do not unintentionally favor some student groups over others. Sensitive student data should be handled responsibly as well as strong privacy protections should be in place.

In conclusion, although there are many obstacles to overcome, improved recommender systems for course selection (CRS) have enormous potential benefits. We can make significant progress toward developing recommender systems that genuinely improve students' educational experiences by filling in the gaps in the present research and using the power of new methods, data and techniques including Artificial Intelegence (AI).

## 9. Conclusions

This study examined 35 research publications that used a variety of recommender system algorithms to suggest academic courses. The most often utilized algorithms were collaborative filtering and novel approaches, with hybrid systems and content-based filtering being employed less frequently. However, each strategy have advantages and disadvantages, and the choice of algorithm depended on several factors, including the availability of data, the goals of the advising system, and use-case context.

This comprehensive SLR covered a total of 35 research papers. Only one study paper utilized the similarity-based filtering technique. On the other hand, thirteen papers used collaborative filtering-based approaches, thirteen papers used novel approaches, six papers used a hybrid recommender systems approach, two papers used content-based filtering approaches, and thirteen papers used novel approaches. The data that was available, the advising system's goals, and the context in which it was being used were some of the factors that affected the recommender system algorithm choice described by these publications.

Although traditional methods were still employed in recent years, the employment of novel methods increased between 2008 and 2020, and some of these methods provided excellent and accurate recommendations. However, a variety of factors, such as the size of the dataset and the number of characteristics employed, affected the outcomes that these approaches produced.

Furthermore, few works discussed preprocessing, and most publications on recommender systems in the field of course suggestions were on the algorithmic level. Data preprocessing is a key step in the process; hence, it must be covered comprehensively by any publication on recommender systems. Numerous publications did not go into adequate detail about the dataset or the implementation's data-splitting technique. Moreover, none of the studies we examined offered to share their data, which makes it very hard to make specific comparisons.

We have drawn the conclusion that collaborative filtering, content-based filtering, hybrid recommender systems, and unique (novel) approaches are all efficient ways to create course recommendation engines based on our SLR and the objectives of the studies that were selected for scrutiny and review. A common technique that concentrates on providing recommendations based on students' prior academic achievement is collaborative filtering (CF). CF has been applied to big data suggestions in education, clustering, and enhancing recall and precision rates. By employing weighted cosine similarity, content-based filtering (CBF) has been utilized to increase the accuracy of recommendations by suggesting courses that are similar to those that students have already taken. Combining the benefits of both CF and CBF, hybrid recommender systems have been used to suggest courses by considering students' learning preferences and learning-objective profiles, as well as to get around collaborative filtering's cold-start issue and CBF's need for domain expertise. Finally, innovative novel techniques have been used for recommending to students the most suitable graduate programs, to customize recommendation systems to each student's estimated prior knowledge background and zone of proximal development, to extract a tailored set of rules from the student elective database, and to develop autonomous course recommender systems for undergraduates using classification models.

Our comprehensive review and findings suggest the following objectives are both desirable and effective for a CRS:

- Making precise course recommendations that are tailored to each student's interests, abilities, and long-term professional goals.
- Addressing the issue of "cold starts," wherein brand-new students without prior course experience might not obtain useful, reliable, and precise advice.
- Ensuring that the system is flexible enough to accommodate various educational contexts, data accessibility, and the unique objectives of the advising system.
- Increasing suggestion recall and precision rates.

- Using preprocessing and data-splitting methods to enhance the predefined performance standards of the CRS overall as well as the predefined and measured quality of recommendations.

Previous research on course recommender systems sheds considerable light on the theoretical foundations of this quickly expanding field. These studies, however, did not analyze real study findings. In the current work, empirical data from studies on course recommender systems are systematically reviewed. We conducted a search of the literature and collected studies that were representative, developed, and included genuine case studies and actual data from the field, and, of course, recommender systems.

The focused evaluation of a few case studies and their findings provided insight into the methodologies used by the various research groups and demonstrated the potential of this new area of educational research. We also identified several gaps that need the researchers' attention along with the opportunities that are now there. This SLR discussed the recent research CRSs to help and improve future research in this area. Researchers need to avoid the weaknesses mentioned herein as well as other literature.

This SLR review emphasizes the value of creating course suggestion tools to help students select classes that are appropriate for their profile (i.e., interests, skill levels, academic predilection and long-term career goals). It is evident from the multiple approaches utilized in the studies under evaluation that creating a successful course recommendation engine necessitates a blend of various strategies and a thorough comprehension of the needs and preferences of the students. Future research in this field can concentrate on incorporating cutting-edge tools such as machine-learning and deep-learning to increase the precision and recall rates of recommendations. Investigating the potential of other data sources to bolster a given learner's profile, from various bases such as social media and Internet-browsing history may create more individualized and precise course recommendation engines.

Generally, there is not a course suggestion system that currently works well for everyone. The most effective system will be determined by the requirements of the students and the information at hand. Future research should, therefore, concentrate on investigating fresh and creative ideas that can get beyond the constraints of current methods and give students more precise and individualized recommendations.

The choice of CRS algorithms in these studies was greatly influenced by the availability of data, the objectives of the advising system, and the use-case(s). Despite our significant observations and results, this study has some drawbacks: (i) only 35 research papers were included in the focused aspect of our study, (ii) the review was hampered by the scant details presented in the included studies, notably regarding the datasets and preprocessing information, (iii) the lack of publicly available CRS datasets prevents fairly comparing the various CRS algorithms. Despite these drawbacks, this study offers a current and thorough analysis of how CRS are functioning and illustrates the directions and next steps this quickly developing discipline should be headed. Additionally, the performance of CRSs should be improved in the future, and recommendations for students should be more individualized, precise and accurate in measurably valid ways.

**Author Contributions:** Conceptualization, S.A. and F.S.; methodology, S.A.; software, S.A.; validation, S.A. and F.S.; formal analysis, S.A.; investigation, S.A.; resources, S.A.; data curation, S.A.; writing—original draft preparation, S.A.; writing—review and editing, S.A. and F.S.; visualization, S.A.; supervision, F.S.; project administration, S.A. and F.S. All authors have read and agreed to the published version of the manuscript.

**Funding:** We acknowledge the support of the Saudi Arabian Cultural Mission (SACM).

**Data Availability Statement:** Data sharing not applicable.

**Conflicts of Interest:** The authors declare no conflict of interest.

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
