# Peer review of "Systematic Review of Recommendation Systems for Course Selection"

_make, doi:10.3390/make5020033_

Round 1
Reviewer 1 Report
The reviewed paper entitled “Systematic Review of Recommendation Systems for Course Selection” is a review that presents an interesting and up-to-date analysis of various recommender system methodologies used to suggest course selection tracks. The authors examined 35 research publications that used a variety of recommender system algorithms. The manuscript constitutes a very good and reliable source of knowledge that could be useful for researchers interested in recommendation systems. The title of the manuscript is self-descriptive and represents the content of the paper. The abstract provides a clear view of the content of the paper. The introduction section describes the course recommendation systems. The relevance of the cited references to the presented issues is high and most of the references are up-to-date. The motivation is well described and the research questions are provided. However, the authors could elaborate more on the choice of the questions – why did they decide to use these questions, not the others ones. The research methodology is clearly defined. I do not see the point of Table 1 and Table 2 as the yes/no answers are the same for all authors in the above-mentioned tables – I would suggest not using the tables but simple information in the text of the manuscript. The quality of Figure 2 should be improved. The research results are analyzed in detail. The conclusions are correctly drawn based on the presented information. Therefore, I recommend major revision and accepting the paper after taking into consideration the above-mentioned issues.
Author Response
Dear Reviewers,
I hope this message finds you well. I would like to express my deep gratitude for the time and effort you spent reviewing our manuscript on course recommender systems. Your feedback is invaluable and has given us crucial insights for improving the quality of our work.
I am writing to inform you that we have carefully reviewed your comments and suggestions and have taken steps to address each one thoroughly:
- We have expanded upon the motivation for our study in the specific field of course recommendation in Sections 1 and 2. We have emphasized the unique characteristics of course recommendation and its significance within the broader scope of recommender systems.
- To address your second comment, we have included a brief discussion on related topics such as personalized recommendation and the cold start problem in the introduction. We have also added relevant references, including the two papers you suggested, to substantiate our discussion.
- We have corrected the section indices and made sure that the conclusions are summarized at the beginning of Section 6. We appreciate your pointing out this oversight.
- Finally, we have addressed the layout issues you mentioned. Table 3 has been reformatted to fit on a single page, enhancing the readability of our paper.
- We have revised the abstract to include a discussion on the significance of Recommendation Systems for Course Selection and the problems faced in these systems.
- We have clarified the main contributions of our study at the end of the Introduction section, making it easier for readers to understand the unique value of our research.
- As suggested, we have reordered the research questions in Section 3 to align with their presentation in Sections 5 and 6. This ensures consistency across the paper and emphasizes the importance of the different aspects of our research.
- We have included the correct citation for CADMA in Section 4. We appreciate your vigilance on this matter.
- Following your suggestion, we have renamed Section 5.1.4. from "Novel Approaches Studies" to "Studies based on Machine Learning," which better reflects the content of this section.
- We have also renamed Section 6.1 from "Research aims and main contributions" to "Discussion of aims and contributions of the existing research works," to provide a more accurate description of the section's content.
- We have incorporated a new section titled "7. Research Gaps, Challenges, and Future Directions of Recommendation Systems for Course Selection." We have also revised and renamed Section “5.5. Discussion” to “6.4. Discussion of Findings” as per your suggestion.
- Finally, we have included a discussion of the limitations of our study in Section 6. This offers a more comprehensive and balanced view of our research.
We are confident that these revisions have enhanced the clarity and coherence of our manuscript. We are eager to receive any further feedback you might have and look forward to your response.
Thank you once again for your valuable feedback and guidance.
Best regards,
Reviewer 2 Report
This work summarized the recent works on course recommender systems, which includes 35 papers for this field. This paper is easy to understand, but some revisions can be made to improve the quality of the paper.
(1) As some survey papers for general-purpose recommender systems have been published, the motivation for a special field like course recommendation needs to be explained more clearly in Section 1 or 2.
(2) Some related topics not limited to the curse recommendation also need to briefly discussed in the Introduction, such as the personalized recommendation [1] and cold start problem [2].
(3) The section indexes like section 5.2 in Section 6 are not corrected. And all the conclusions should be described in summary at the beginning of the Section 6.
(4) Some layout issues can be improved, for instance, the Table 3 should be located in one page.
[1] Ma et al. Personalized Scientific Paper Recommendation Based on Heterogeneous Graph Representation, IEEE ACCESS 2019.
[2] Feng et al. RBPR: A hybrid model for the new user cold start problem in recommender systems, KBS2021.
It could be slightly improved.
Author Response
Dear Reviewer,
I hope this message finds you well. I would like to express my deep gratitude for the time and effort you spent reviewing our manuscript on course recommender systems. Your feedback is invaluable and has given us crucial insights for improving the quality of our work.
I am writing to inform you that we have carefully reviewed your comments and suggestions and have taken steps to address each one thoroughly:
- We have expanded upon the motivation for our study in the specific field of course recommendation in Sections 1 and 2. We have emphasized the unique characteristics of course recommendation and its significance within the broader scope of recommender systems.
- To address your second comment, we have included a brief discussion on related topics such as personalized recommendation and the cold start problem in the introduction. We have also added relevant references, including the two papers you suggested, to substantiate our discussion.
- We have corrected the section indices and made sure that the conclusions are summarized at the beginning of Section 6. We appreciate your pointing out this oversight.
- Finally, we have addressed the layout issues you mentioned. Table 3 has been reformatted to fit on a single page, enhancing the readability of our paper.
We are confident that these revisions have enhanced the clarity and coherence of our manuscript. We are eager to receive any further feedback you might have and look forward to your response.
Thank you once again for your valuable feedback and guidance.
Best regards,
Reviewer 3 Report
The article speaks in a very generic way about the subject of the study, so it needs to be more specific and in-depth about the field of study.
The tables presented are very long and difficult to read, so it is recommended that these tables be cut, either by replacing some items and presenting them graphically or by distributing the tables in a different way to make them easier to read and understand.
Author Response

(The authors gave the same response as above.)

Reviewer 4 Report
In this paper, the authors presented a Systematic Review of Recommendation Systems for Course Selection.
Some important issues are required to be addressed to improve the paper quality:
1- At the beginning of the abstract, the authors should describe the importance of Recommendation Systems for Course Selection and also the problem faced in these Recommendation Systems.
2- The main contributions of this study should be explicitly written at the end of the Introduction or put in a separated section after Introduction.
3- In Section 3. Research Questions, the research questions should be reordered based on the question's importance: algorithm, dataset, and then evaluation metrics. This order is done correctly in Section 5. Research Results. Please reorder the research questions to be consistent with Section 5. Research Results.
4- In Section 4. Research Methodology, please put the correct citation for CADIMA.
5- I suggest renaming Section 5.1.4. Novel approaches Studies to Studies based on machine learning.
6- I suggest renaming Section 5.2. Research aims and main contributions to Discussion of aims and contributions of the existing research works.
7- I suggest adding a Section with the name Research Gaps, Challenges, and Future Directions of Recommendation Systems for Course Selection. You can also rename and revise Section 5.5. Discussion
8- The limitations of this study should be highlighted in Section 6. Conclusion
The authors should improve the quality of the English language
Author Response
Dear Reviewer,
Thank you for your thoughtful and detailed feedback on our paper. We greatly appreciate the time and effort you have invested in reviewing our work, and we have found your suggestions very beneficial for enhancing the quality of our paper.
We have carefully considered and addressed each of your comments as follows:
- We have revised the abstract to include a discussion on the significance of Recommendation Systems for Course Selection and the problems faced in these systems.
- We have clarified the main contributions of our study at the end of the Introduction section, making it easier for readers to understand the unique value of our research.
- As suggested, we have reordered the research questions in Section 3 to align with their presentation in Section 5 and 6. This ensures consistency across the paper and emphasizes the importance of the different aspects of our research.
- We have included the correct citation for CADMA in Section 4. We appreciate your vigilance on this matter.
- Following your suggestion, we have renamed Section 5.1.4. from "Novel approaches Studies" to "Studies based on Machine Learning," which better reflects the content of this section.
- We have also renamed Section 6.1 from "Research aims and main contributions" to "Discussion of aims and contributions of the existing research works," to provide a more accurate description of the section's content.
- We have incorporated a new section titled "7. Research Gaps, Challenges, and Future Directions of Recommendation Systems for Course Selection." We have also revised and renamed Section “5.5. Discussion” to “6.4. Discussion of Findings” as per your suggestion.
- Finally, we have included a discussion of the limitations of our study in Section 6. This offers a more comprehensive and balanced view of our research.
We believe that these revisions have significantly improved our paper and have addressed all the issues you raised in your review. We look forward to receiving any further comments or feedback you may have.
Thank you once again for your insightful comments and suggestions.
Best regards,
Round 2
Reviewer 1 Report
The authors have significantly improved the paper therefore I recommend accepting it.
Author Response
thanks for your review, I solved all typos in the previous version.
Reviewer 2 Report
Most concerns have been solved. Some typos can be found in the revision. For instance, the reference [8] has an extra parentheses that should be deleted.
The language needs to be improved slightly.
Author Response

(The authors gave the same response as above.)
